# Diagnostics of HNSCC Patients: An Analysis of Cell Lines and Patient-Derived Xenograft Models for Personalized Therapeutical Medicine

**DOI:** 10.3390/diagnostics12051071

**Published:** 2022-04-25

**Authors:** Ramona Gabriela Ursu, Ionut Luchian, Costin Damian, Elena Porumb-Andrese, Nicolae Ghetu, Roxana Gabriela Cobzaru, Catalina Lunca, Carmen Ripa, Diana Costin, Igor Jelihovschi, Florin Dumitru Petrariu, Luminita Smaranda Iancu

**Affiliations:** 1Department of Microbiology, Faculty of Medicine, “Grigore T. Popa” University of Medicine and Pharmacy, 700115 Iasi, Romania; ramona.ursu@umfiasi.ro (R.G.U.); costin.damian@d.umfiasi.ro (C.D.); cobzaru.roxana@umfiasi.ro (R.G.C.); catalina.lunca@umfiasi.ro (C.L.); ripa.carmen@umfiasi.ro (C.R.); diana.costin@umfiasi.ro (D.C.); jelihovschi.igor@umfiasi.ro (I.J.); luminita.iancu@umfiasi.ro (L.S.I.); 2Department of Periodontology, Faculty of Dental Medicine, “Grigore T. Popa” University of Medicine and Pharmacy, 700115 Iasi, Romania; 3Department of Dermatology, Faculty of Medicine, “Grigore T. Popa” University of Medicine and Pharmacy, 700115 Iasi, Romania; elena.andrese@umfiasi.ro; 4Department of Plastic Surgery, Faculty of Medicine, “Grigore T. Popa” University of Medicine and Pharmacy, 700115 Iasi, Romania; ghetu.nicolae@umfiasi.ro; 5Department of Hygiene and Environmental Health, Faculty of Medicine, “Grigore T. Popa” University of Medicine and Pharmacy, 700115 Iasi, Romania; florin.petrariu@umfiasi.ro

**Keywords:** HNSCC, HPV, PI3K, cell lines, patient-derived xenograft models

## Abstract

Head and neck squamous cell carcinomas (HNSCC) are very frequent worldwide, and smoking and chronic alcohol use are recognized as the main risk factors. For oropharyngeal cancers, HPV 16 infection is known to be a risk factor as well. By employing next-generation sequencing, both HPV-positive and negative HNSCC patients were detected as positive for PI3K mutation, which was considered an optimal molecular target. We analyzed scientific literature published in the last 5 years regarding the newly available diagnostic platform for targeted therapy of HNSCC HPV+/−, using HNSCC-derived cell lines cultures and HNSCC pdx (patient-derived xenografts). The research results are promising and require optimal implementation in the management of HNSCC patients.

## 1. Introduction

HNSCC represents a heterogenous group of tumors, including cancer of the oropharynx, oral cavity, pharynx, and larynx. The recognized risk factors for HNSCC are smoking, chronic alcohol use, lack of oral hygiene, and HPV 16 in oropharyngeal cancers. Recent meta-analyses have confirmed smoking as a risk factor for HNSCC: Alotaibi et al. found that smoking is a negative prognostic factor for overall survival in patients with hr-HPV^+^ [1], and Ference et al. found that current smoking during treatment is associated with the greatest reduction in survival [2]. Interestingly, Skoulakis et al. found in their meta-analysis that smoking is less common in HPV-positive groups than in HPV-negative groups [3]. The role of HPV in HNSCC was confirmed by a meta-analysis which included 148 studies and 12,163 cases of HNSCC from 44 countries, and the authors found HPV16 was present in more than 80% of all HPV DNA-positive cases [4]. Updated information regarding incidence, prevalence and mortality are available on the Cancer Today website by the International Agency for Research on Cancer (IARC). The estimated age-standardized incidence rates, in 2020, for lip, oral cavity, oropharynx, nasopharynx, hypopharynx cancers, both sexes, all ages, were 7.4 in USA, 12.7 in France, 12.9 in Romania, 6.5 in Brazil, 4.8 in China, 9.0 in Namibia and 9.8 in Australia [5]. The estimated numbers of prevalent cases (5-year) as a proportion in 2020 for hypopharynx, lip, oral cavity, nasopharynx, and oropharynx (both sexes, all ages) were 39.2 in USA, 65.9 in France, 59.6 in Romania, 20.6 in Brazil, 20.1 in China, 12.0 in Namibia and 51.0 in Australia [6]. The estimated age-standardized mortality rates (World) in 2020 for hypopharynx, lip, oral cavity, nasopharynx, and oropharynx (both sexes) all ages were 1.4 in USA, 3.0 in France, 6.7 in Romania, 3.2 in Brazil, 2.5 in China, 5.5 in Namibia and 1.6 in Australia [7].

CANCER TOMORROW, another project of IARC, enables a quantification of the future cancer burden changes of new cases from 2020 to 2040, both sexes, all continents, age (0–85+): hypopharynx 46.9%, lip, oral cavity 41.7%, nasopharynx 37.3%, oropharynx 35.5%. For Romania, the highest changes of new cases from 2020 to 2040, both sexes, age (0–85+) are predicted to be for lip, oral cavity (6.7%) and hypopharynx (2.8%) [8].

A recent multicenter study concluded that in some populations in the United States, more than 90% of OPSCCs are produced by HPV [9]. The updated data available on the International Agency for Research on Cancer (IARC) Cancer Today website underline the importance of this health issue and raise some questions regarding the risk factors for developing different types of head and neck squamous cell carcinoma (HNSCC), depending on age and gender. HPV 16 was recognized as a risk factor for oropharyngeal cancers, besides smoking and chronic alcohol use [10]. In a recent meta-analysis, Mariz Bala et al. analyzed the data of more than 6000 patients to reveal accurate information about the global prevalence of human papillomavirus (HPV) in oropharyngeal squamous cell carcinomas (OPSCC). Compared to the overall HNSCC prevalence, which is different for male and females, the authors identified a similar 45% pooled prevalence of HPV-driven OPSCC, for both genders, and they also suggested that double p16/HPV-DNA/RNA testing is the optimal method in regard to specificity and prognostic accuracy [11].

The treatment of HNSCC includes surgery, chemotherapy, and radiotherapy alone or combined. In some cases, resistance to therapy with recurrences and metastasis and/or side effects appear. This underlines the need for a new direction of research for targeted cancer therapy. Phosphoinositide 3-kinase (PI3K)/mammalian target of rapamycin (mTOR) pathway components are key therapeutic targets in cancer, immunity, and thrombosis. In normal cells, the PI3K/mTOR pathway has regulatory roles in cell survival, proliferation, and differentiation. However, aberrant variants of activation of this pathway frequently occur in human cancers [12]. PI3K is believed to be one of the key therapeutic targets for cancer treatment, based on the observation that hyperactivity of PI3K signaling is significantly correlated with human tumoral progression, an increase in tumor micro vessel density and enhanced chemotaxis and invasive potential of cancer cells. Enormous efforts have been dedicated to the development of drugs targeting PI3K signaling, many of which are currently employed in clinical trials evaluation. PI3K inhibitors are subdivided into dual PI3K/mTOR inhibitors, pan-PI3K inhibitors and isoform-specific inhibitors [13].

The most used inhibitors in the treatment of solid tumors are the pan-PI3Kis (Buparlisib—BKM120; Pictilisib—GDC-0941 and Copanlisib—BAY 80-6946), which target each of the four catalytic isoforms of class I PI3K; therefore, they have the potential for broad activity in several tumors types, with a range of different molecular alterations. However, such broad inhibition of this molecular pathway may lead to a potentially higher risk of adverse events, which could limit the use of such agents in therapeutic doses. BEZ235 is a potent, oral, ATP-competitive dual inhibitor of the four class I PI3K isoforms and the downstream effectors mTORC1/2. Alpelisib—BYL719 isoform-specific PI3Kis have the narrowest profile and may require careful patients’ selection based on potential biomarkers of sensitivity and resistance [12].

The novelty of these targeted therapies meant that besides having the promise of potentially serving as new treatment strategies, several lessons had to be learned from early studies. The findings to date suggest that PIK3CA and PTEN alterations are relatively weak biomarkers of clinical activity. However, PIK3CA mutations appear to be more promising as predictive factors for p110a catalytic isoform-specific inhibitors, with PTEN alterations possibly associated with resistance. Secondly, it is increasingly evident that single-agent targeting of the PI3K pathway has limited activity. Therefore, the identification of appropriate biomarkers of efficacy and the development of optimal combination therapies and dosing schedules for PI3Kis are likely to be required for the broad acceptance of this class of compounds in clinical practice [13].

The acquired amplification and mutation of PIK3CA and PIK3CB, which resulted in a marked upregulation of the PI3K signaling itself, has been shown to cause resistance to selective PI3K inhibitors [12].

Overall, PI3K inhibition is being investigated as a potential strategy to develop novel therapeutics for cancer management. Although different researchers are moving forward with the clinical development of PI3K inhibitors, maximizing the utility of these agents in the treatment of cancer patients remains challenging. Certainly, understanding the precise mechanisms of PI3K signaling and PI3K inhibition will be critical. Optimization of the patient selection strategies and combination approaches will help increase the practical efficacy of these agents. Continued work to clarify the resistance mechanisms and the novel strategies to overcome resistance will also be important [12].

### HNSCC Patients PI3K Inhibitors Clinical Trials

Over the last 5 years, five clinical trials were published (three from the USA, one from Canada and one from France), which evaluated the PI3K targeted therapy in recurrent or metastatic HNSCC patients, heavily pre-treated HNSCC patients, or locoregionally advanced SCCHN (LA-SCCHN) patients.

Chronologically, the clinical trials analyzed the synergistic effects of the combination of temsirolimus with low-dose weekly carboplatin and paclitaxel [14]; assessed the maximum tolerated dose (MTD) of the PI3K inhibitor buparlisib given concurrently with cetuximab [15]; evaluated the addition of BYL719 to cetuximab and radiation [16]; and assessed the effects of alpelisib, a class I α-specific PI3K inhibitor in combination with concurrent cisplatin-based chemoradiation [17] and a combination of copanlisib, an intravenous, pan-class I PI3K inhibitor, with the anti-EGFR monoclonal antibody cetuximab [18]. Tumor regressions and benefit from the given PI3K therapy was reported for combining mTORC1 inhibitors with carboplatin and paclitaxel chemotherapy, buparlisib at 100 mg daily plus cetuximab, BYL719 associated with cetuximab and radiation, Alpelisib in combination with cisplatin-based CRT (where the three-year overall survival was 77.8%), and Axitinib, a potent inhibitor of vascular endothelial growth factor receptor [14,15,16,17,19]. The most recent trial [18] studied the novel drug copanlisib combined with cetuximab and demonstrated unfavorable toxicity and limited efficacy, and the trial was stopped earlier than initially planned.

It is more than obvious that nowadays, there is a growing amount of important research and discoveries in the field of developing specific inhibitors and in the field of technology for assessing the efficiency of these cell lines treatment with specific inhibitors. At the same time, medical specialties develop practices separately from other specialties, sometimes without taking into consideration the discoveries of other medical fields. The results obtained from laboratory should be transmitted and applied in clinical practice for the optimization of the cancer patients’ follow-up. For example, it would be necessary to know the cytotoxic effect needed for each patient. With selected antibiotics, it is possible to determine the lowest dose of antibiotic needed to kill a bacterium. We might try to come up with a similar approach for tumor target therapy, as for the moment, it is not sure if oncologists measure the levels of anti-tumoral drugs, while the oncologic patients continue therapy despite moderate side effects. For example, patients could have to tolerate pneumonitis and leg edema from high doses of everolimus. One needs uninterrupted, high levels of a certain drug when using prolonged therapy (antibiotic or likely anticancer); resistance develops with stops/starts or lower doses. Another important aspect to be taken into consideration is if the absorption of oral drugs may be affected by food intake. To delay or avoid resistance, we might have to use a combination of multiple drugs that attack the same target in a similar matter to how we avoid resistance when administering antibiotics, by combining two drugs that act on the same target, e.g., the cell wall. By looking at how other medical specialties deal with similar negative outcomes, such as resistance and establishing minimum effective doses, we may develop better strategies for the treatment of cancer patients [20].

Aim: to analyze the availability, sensitivity and specificity of the new diagnostic platform for targeted therapy of HNSCC HPV+/−, using HNSCC-derived cell lines culture and HNSCC pdx (patient-derived xenografts).

## 2. Materials and Methods

Literature search and study selection: a systematic search of the PubMed and the EMBASE databases was carried out for all the published studies on HNSCC in the last 5 years, using the following search algorithm: HNSCC HPV-positive PI3K-positive targeted therapy, HNSCC cell lines PI3K-targeted therapy, pdx xenograft HNSCC HPV PI3K-targeted therapy. We performed a systematic analysis for the studies that were published in English, from 1 January 2017 to 1 March 2022, and that described and analyzed the methods used for optimal targeted therapy of HNSCC patients. The 14 studies were blinded and analyzed by two persons (Figure 1). We excluded review papers and studies that have tested cell lines to other therapies except PI3K inhibitors.

## 3. Results

a. We identified five studies that approached different HPV-positive or HPV-negative HNSCC cell lines, with or without PI3K mutation, and tested the effects of different PI3K inhibitors, alone or in combination with other drugs (e.g., cisplatin and docetaxel). In addition to PI3K, the authors identified other molecular targets, such as HRAS and HER3. All these studies detected that the tested HNSCC cell lines were sensitive to the selected drugs, and they suggested the continuation of these studies provides a rationale for the clinical evaluation of targeted therapy for the treatment of HPV+ HNSCC patients. Therapeutical effect was evaluated using specific and sensitive diagnostic methods, including evaluation of viability, proliferation, cytotoxicity, and apoptosis [21,22,23,24,25] (Table 1).

b. We identified nine studies that used different pdx HPV-positive HNSCC models. The majority of the analyzed studies were performed in the USA, and all authors looked for the successful establishment of pdx models from HNSCC, including preserving the genomic profile (e.g., HPV status, p16, PI3K). The established pdx were treated with specific drugs: PI3K inhibitors alone or in combination with cetuximab, pan-HER inhibitors, and Spleen tyrosine kinase (SYK) inhibitors. The results of this research clarify the basic profile of HNSCC, the molecular mechanisms of resistance to the treatment, and of course, their potential for the development of novel molecular therapy [26,27,28,29,30,31,32,33,34] (Table 2). The available clinical characteristics of patients from which the pdx were derived can be seen in the Table 3.

## 4. Discussion and Conclusions

In our descriptive literature review, we have analyzed the recent studies that evaluated the applicability of two modern platforms of diagnostics: cell lines and pdx derived from HNSCC and their response to PI3K inhibitors. In the last 5 years, comprehensive studies were published, which focused on the preparation and validation of these diagnostic platforms. Validation was achieved by preserving the histology and genomic profiling of the original tumors. Most of the tested cell lines were sensitive to the PI3K drugs, but a synergistic effect was seen in case of the association between novel targeted therapy together with known oncologic treatment or with inhibitors of other molecular targets.

HNSCCs are different regarding HPV involvement as an etiologic factor in comparison to cervical cancer. If for cervical cancer there are very well-established guidelines to select women that present a high risk for carcinogenesis, progression, and invasion [35], and there is hope for the eradication of cervical cancer [36], in the case of HNSCC, only one single high risk HPV type, 16, is recognized as a risk factor, and only for OPSCC (oropharyngeal cancers), beside smoking and chronic alcohol use [10].

For HNSCC, the current standard of care, for most patients with head and neck squamous cell carcinoma, remains a combination of surgery, radiation and/or cytotoxic chemotherapy [37].

One research direction is to identify the HPV-driven HNSCC cases by using a very strict algorithm of diagnosis, similar in manner to the HPV-AHEAD project, which was realized by an international team of researchers, under the guidance of Infections and Cancer Biology Group, International Agency for Research on Cancer, Lyon, France. The algorithm used in this study was a very strict and rigorous one: FFPE HNSCC samples were analyzed using HPV DNA testing, HPV RNA testing, and p16 analysis. Archived HNSCC tissue samples—189 from northeastern Romania, 364 from the central region of India, 696 from Italy and 772 patients from Belgium—were tested using this algorithm. In all the four HPV-AHEAD studies, the highest rate of HPV DNA prevalence was detected for OPSCCs, with similar values in the studied areas (50% in Romania, 18.9%% in India, 40.4% in Italy and 36.4% in Belgium). HPV 16 was the most prevalent viral type in all the samples analyzed by these studies. HPV-driven HNSCCs were defined by the presence of both viral DNA and RNA, and the highest prevalence of this double positivity was also found in OPSCC samples [38,39,40,41]. The utility of detection of HPV in HNSCC cases will lead to the opportunity to prevent these cancers by available HPV vaccines.

Another research direction on HNSCC is to identify predictive biomarkers or targetable mutations, employing the use of advances in precision medicine, e.g., next-generation sequencing (NGS). HPV-positive oropharyngeal cancers have a better clinical outcome than HPV-negative cases when given radiotherapy (RT) alone and subsequent surgery if needed [42]. Most HPV-positive HNSCC patients may not need intensified chemotherapy or hyper fractionated radiotherapy, and less intensive treatment would be a better option, in order to avoid side effects. The patients who are identified as HPV positive can benefit from a de-escalation of therapy; thus, more specific diagnosis assays can be applied directly in clinical practice. In a Swedish study, when hotspot mutations in 50 cancer-related genes were analyzed by NGS, PI3KCA and FGFR3 mutations were frequently detected in HPV-positive but not in HPV-negative TSCC/BOTSCC [43]. Continuation and complementary studies were realized for patients with HPV + TSCC/BOTSCC and wild-type FGFR3, and researchers found that the overexpression of FGFR3 was correlated with better disease-free survival (DFS) [44,45,46].

These findings are supported by recent studies, such as one published in March 2022: PIK3CA gene mutations were present in almost 40% of HNSCC samples, and the authors considered that these patients could benefit from therapies targeting the PI3K pathway, using further methodological standardization [47].

Previous research (2018) from Queensland, Australia considered that PIK3CA mutations may serve as predictive biomarkers for therapy selection. Therefore, the authors developed an allele-specific technology for the detection of PIK3CA alterations in circulating tumoral DNA (ctDNA). ctDNA holds promise as a potential biomarker in HNSCC [48]. Janecka-Widła et al. identified differences regarding the expression and prognostic potential of proteins involved in PI3K signaling between HPV 16-positive and HPV-negative HNSCC patients, using immunohistochemistry and qPCR [49].

In this review, we presented the updated results regarding different diagnostic platforms for guiding targeted therapy of HNSCC, both HPV positive and negative. In addition to the PI3K pathway [22,23], the analyzed studies identified other molecular targets, such as HER3 [24,25] and HRAS [21], which could be effective therapeutic targeting strategies in HNSCC cell lines, either HPV positive or negative. The multiple therapeutical targeting (e.g., TP53, CDKN2A, CCND1, EGF receptor—EGFR) in HPV-positive and negative HNSCC is supported by the findings of other authors [50,51,52].

Patient-derived xenografts employed as models for head and neck cancer are recent and modern platforms to optimize and discover new targeted therapy for head and neck tumors. PDX are offering the opportunity of ”personalized” treatment of HNSCC patients, as they have the ability to predict clinical outcomes of the same patient undergoing [53]. PDX are offering more comprehensive data regarding targeted therapy, in comparison with different HNSCC cell lines, which are monolayer cells. The PDX are offering the possibility to simulate the heterogeneity of clinical HNSCC, with histological, pathological and genetical similarities, and therefore, they are considered relevant models for precision medicine in HNSCC [54].

The promising results here presented, using patient-derived xenografts and patient-derived cells for HNSCC, were validated in a recent review regarding their preclinical application in evaluating some personalized medicine strategies as a response to the need for new targeted therapy of HNSCC [55,56].

Genomic alterations and key pathways involved in the formation of HNSCC and the clinical presentation. Recently, many groups of researchers are focusing on the identification of new possible therapeutic targets in HNSCC. Chen et al. (2021) identified four novel HNSCC susceptibility loci (CDKN1C rs452338, CDK4 rs2072052, E2F2 rs3820028 and E2F2 rs2075993) as genetic alterations in the cell cycle pathway that are common in HNSCC [57]. Zhang et al. analyzed for the first time the 2-hydroxyisobutylated modification proteomic for OSCC, which is significantly concentrated in the actin cytoskeleton regulatory pathway, suggesting that this pathway may mediate the oncogenesis or exacerbation of OSCC [58]. Li et al. studied the role and molecular mechanism of cyclin-dependent kinase 5 (CDK5) in regulating the growth of tongue squamous cell carcinoma (TSCC). The authors found that increased levels of CDK5 expression in TSCC tissues was established as an independent risk factor affecting TSCC growth and patient prognosis. CDK5 was proven as an oncogene in TSCC and is considered as a molecular marker for use in the diagnosis and treatment of TSCC [59]. Another research team identified a promising therapeutic target and prognosis marker for human OSCC: actin-like protein 8, which was detected to play an oncogenic role in the pathogenesis of OSCC [60]).

Screening and prevention of HNSCC. Many research studies are focusing on identification of the optimal biomarker for the early detection of HNSCC, as the routine clinical evaluation includes clinical examination and radiological assessment. Huang et al. identified in a meta-analysis that circular RNAs showed high accuracy in the diagnosis of OSCC and could be used as prospective biomarkers for optimal diagnostic [61]. Another possible biomarker for early detection is hypermethylated DNA in saliva and oral swabs for OSCC, which proved to have optimal accuracy and raised hope for the optimization of fast detection of these cancers’ evolution, given its non-invasive sampling procedures [62]. Gaw et al. supported non-invasive hypermethylation markers using saliva and oral swabs for OSCC diagnosis, together with other biomarkers, to optimize the sensitivity and specificity of screening [63]. One possibility to prevent HPV-associated HNSCC is HPV vaccination program implementation for males also [64].

Many studies are analyzing the implications of PI3K pathway alteration for the EGFR pathway in HNSCC. Zaryouh et al. are considering that co-targeting EGFR and the PI3K/Akt pathway could have a synergistic drug effect, improving sensitivity to EGFR and clinical efficacy. Optimal selection is important for patients who could benefit from this targeted therapy [65]. The anti-EGFR monoclonal antibody cetuximab was evaluated in recurrent and/or metastatic HNSCC patients in a phase I dose-escalation trial, and the authors stopped the trial because of the unfavorable toxicity profile [18].

Genetic alterations in epidermal growth factor receptor (EGFR) and PI3K are common in HNSCC: Mock et al. found that more than half of HPV-negative HNSCC showed a pathway activation in EGFR or PI3K [66].

A clinical trial published in Lancet Oncology by Soulières et al. evaluated if the addition of buparlisib to paclitaxel could optimize clinical outcomes compared with paclitaxel and placebo in patients with recurrent or metastatic HNSCC. After 2 years of follow up of more than 150 patients, the authors observed improved clinical efficacy with a manageable safety profile, and they suggested that buparlisib in combination with paclitaxel could be an effective second-line treatment for recurrent or metastatic HNSCC patients [67]. One year later, the same research team published another clinical trial in which patients with TP53 alterations, HPV-negative status, and low mutational load had a better overall survival from combination therapy of buparlisib and paclitaxel [68]. Brisson RJ et al. evaluated on 12 patients the maximum tolerated dose of buparlisib given concurrently with cetuximab in recurrent and metastatic HNSCC. In this pilot study, the authors concluded that buparlisib at 100 mg daily plus cetuximab proved to be well-tolerated [15]. Lenze et al. considered that buparlisib, a class I pan-PI3K inhibitor, with an optimal tolerated toxicity, could improve the actual 5-year overall survival for HNSCC, which is only at around 50–66% [69].

Some group researchers have used PDX formation to predict the increased risk to HNSCC. Facompre et al. established well-characterized PDXs and organoids from HPV+ HNSCCs, and they proved the retaining of PIK3CA mutations, TRAF3 deletion, and the absence of EGFR amplifications and NOTCH1 mutations. The authors have identified, in a PDX model, reduced E7 and p16INK4A levels, which were associated with recurrent HPV+ HNSCCs and with lethal outcome. For the prediction of disease recurrence risk, the authors have also analyzed on PDX the E2F target gene expression as a useful biomarker [27]. PDX models have been used to predict the risk to HNSCC and, in an evaluation of potential biomarkers, such as Remodelin, an inhibitor of NAT10 (one of the most promising prognostic risk gene). In their study, the authors demonstrated it significantly suppressed the growth of HNSCC in a PDX model, indicating that Remodelin may be an optimal candidate drug for HNSCC treatment [70]. In a very recent study, the authors aimed and succeeded at developing a prediction score for locoregional failure and distant metastases in OSCC that incorporates PDX engraftment, beside the known clinicopathological risk factors [71].

In a comprehensive study, Karamboulas et al. have studied the molecular profiling of 64 engrafters and 48 non-engrafters, which were tested for DNA mutations or a number of copies alterations with a custom hybrid capture targeted sequencing panel of 112 genes. The authors found no statistically significant associations between any single gene mutation and engraftment. They identified that CDKN2A mutation or deletion, a CCND1 amplification, or both were present in 83.4% in case of rapid engrafters, while there were only 18.2% in case of non-engrafting/slow engrafters, which suggest that genomic deregulation of the G1/S checkpoint pathway correlates with engraftment [72]. Using the Illumina Cancer Hotspot Panel, Strüder et al. analyzed the molecular profile (TP53, KDR, KRAS, SMARCB1, EGFR) of 13 patient samples and corresponding PDX, and they found that molecular pathology is preserved in PDX, with the mention regarding the importance of intratumoral heterogeneity [26].

An impressive multidisciplinary research team from the Princess Margaret Cancer Centre, Toronto, Canada, raised the level of using PDX models from recapitulating many of the features of their corresponding clinical cancers, including histopathological and molecular profiles, to preclinical assessment of CDK4 and CKD6 inhibition, using abemaciclib, on a large collection of 243 HPV-negative HNSCC patient-derived xenograft models. The authors are underlying the necessity of using this type of CDK4 and CDK6 inhibitors in case of HNSCC patients with the poorest prognosis, and even more, they are mentioning the necessity of further studies for the identification of mechanisms of resistance of abemaciclibin, alone or in association with ionizing radiation [72].

Our study is a descriptive review of the recent findings regarding the latest research studies that are focusing on the evaluation of the most optimal targeted therapy of HNSCC, using cell lines and HNSCC-derived pdx as models. As a future research direction, we will be developing patient-derived cancer organoids that bridge the conventional gaps in PDC and PDX models [73]. In addition, a more important approach will be to assist clinicians on how to appropriately incorporate all this preclinical research in the optimal management of HNSCC [37].

## Figures and Tables

**Figure 1 diagnostics-12-01071-f001:**
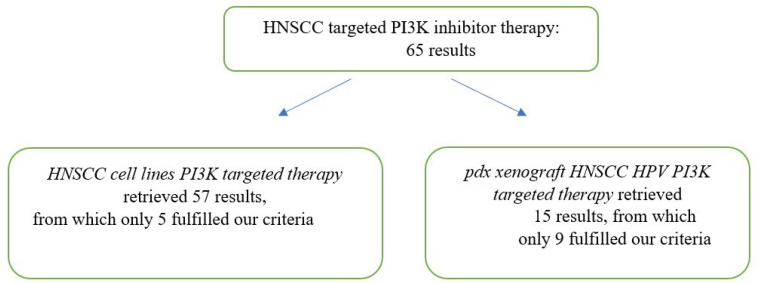
Flow chart of the analyzed studies.

**Table 1 diagnostics-12-01071-t001:** Diagnostic evaluation of HNSCC cell lines with targeted PI3K inhibitors therapy.

First Author Year, Country	Type of Cell Lines	Targeted Therapy	Results	Novelty
Javaid et al., 2022 United States [21]	HRAS-mutant HNSCC cell lines HPV negative	the activity of tipifarnib and inhibitors of HRAS effector signaling	concurrent inhibition of HRAS effector signaling (PI3K) increased sensitivity to tipifarnib	further research is needed to apply these combinations and others for HRAS-mutant HNSCC
Holzhauser et al., 2021 Stockholm, Sweden [22]	HPV+ CU-OP-2, -3, -20, UPCI-SCC-154, and HPV- CU-OP-17 and UT-SCC-60A cell lines	alpelisib (BYL719) and erdafitinib (JNJ-42756493) alone and in combination with cisplatin or docetaxel	- dose-dependent responses to all PI3K/FGFR inhibitors - cisplatin and docetaxel induced dose dependent responses	alpelisib and erdafitinib were efficient in inhibiting TSCC/BOTSCC cell line growth
Holzhauser et al., 2019, Stockholm Sweden [23]	two HPV+ and one HPV- TSCC/BOTSCC cell lines	FGFR inhibitor AZD4547 + PI3K inhibitors BEZ235 and BKM120 were tested alone on HPV + UM-SCC-47, and UPCI-SCC-154	all cell lines were found sensitive to treatment with these drugs	this underlines the need for further investigations of TSCC/BOTSCC cell lines with FGFR3 and PIK3CA mutations
Brand et al., 2018 San Francisco California [24]	HPV (+) HNSCC cell lines	the efficacy of PI3K-targeted therapies	HER3 and PI3K may be an effective therapeutic strategy in HPV (+) tumors	this drug combination may improve patient outcomes in HPV (+) head and neck cancers
Brand et al., 2017 San Francisco CA, USA [25]	HPV+ HNSCC cell lines	HER3 was investigated as a molecular target HER3 was overexpressed in HPV+ HNSCC	targeting HER3 with siRNAs or KTN3379 significantly inhibited the growth of HPV+ cell lines	HER3 is directly related to HPV infection in HNSCC

**Table 2 diagnostics-12-01071-t002:** Diagnostic evaluation of pdx HNSCC with targeted PI3K inhibitors therapy.

First Author Year, Country	PDX Type	PDX Assessing	Results	Novelty
Strüder et al., 2021, Rostock, Germany [26]	55 HNSCC samples fresh frozen and implanted NOD.Cg-PrkdcscidIl2rgtm1Wjl/SzJ mice	PDX-take rate of growth, histopathology, molecular characteristics	High PD-L1 expression (combined positive score on tumor/immune cells) predicted PDX rejection. The p16 status had no impact on engraftment efficacy.	This study presents the successful establishment of patient-derived xenograft models obtained by endoscopic biopsy resections of head and neck tumors.
Facompre et al., 2020 Philadelphia, Pennsylvania [27]	panel of patient-derived xenografts (PDXs) and organoids from HPV+ HNSCCs	Novel association between tumor mutational burdens (TMBs) and local progression in both HPV+ and HPV- patients	Reduced E7 and p16INK4A levels found in a PDX from an outlier case with lethal out-come led to the detection of similar profiles among recurrent HPV+ HNSCCs.	Critical gap in preclinical models for HPV+ HNSCCs focuses on new potential applications of the functions of viral oncogenes for biomarker development.
Klinghammer et al., 2020 Berlin, Germany [28]	mouse clinical trial set-up with 33 PDX models with known HPV and PI3K mutational status and available data on cetuximab sensitivity	The antitumor activity of the selective, pan-class I PI3K inhibitor copanlisib in monotherapy and in combination with cetuximab	Treatment with copanlisib alone resulted in moderate antitumor activity with 12/33 PDX models showing either tumor stabilization or regression. Combination treatment with copanlisib and cetuximab was superior to either of the monotherapies alone in the majority of the models (21/33), and the effect was particularly pronounced in cetuxi-mab-resistant tumors (14/16).	This study underlines the importance of PI3K inhibition in HNSCC and considers gene expression patterns as a promising biomarker for predicting the response to treatment.
Kang et al., 2020 Kyungbuk South Korea [29]	HNSCCs 15/62 (24%) PDXs were established	Established PDXs by histology, genomic profiling, and in vivo anti-cancer efficacy testing to confirm them as the authentic in vivo platform	PDXs mostly retained donor characteristics and remained stable across passages. PIK3CA, HRAS, and TP53 mutations and EGFR, CCND1, MYC, and PIK3CA amplifications were identified.	The response of PDX to the drugs was very similar to the response of donor patients who were treated with pan-HER and pan-PI3K inhibitors.
Black et al., 2019 Ontario, Canada [30]	panel of 28 in vivo HNSCC PDX models	The potency of SYK inhibitor ER27319 maleate on cellular prolife-ration	Treatment of PDXs with ER27319 maleate was observed to reduce tumor burden in vivo in two of three models	This study shows that SYK can be a target for HNSCC therapy.
Berggren et al., 2019 Albuquerque, NM [31]	mouse PDX models HNSCC models	Inhibition of MK2 by pharmacologic and genetic methods impacts tumor growth.	- Decreased tumor growth - Increased survival	This study establishes that the MK2 pathway mediates the resistance to radiation treatment while at the same time may serve as a prognostic biomarker.
Lilja-Fischer et al., 2019 Aarhus Denmark [32]	12 squamous cell carcinoma PDX models (7 HPV+, 5 HPV-)	Established PDX models maintained histological and immunohistochemical characteristics as well as HPV status of the primary tumor	Important cancer driver gene mutations, e.g., in TP53, PIK3CA were preserved.	For OPSCC tumors, PDX retains the molecular characteristics of the human primary tumor.
Folaron et al., 2019 NY USA [33]	panel of PDX models of HNSCC and the impact on therapeutic response	PDX models retained the HPV/p16 status of the original patient tumor	The PDX with the tumor vessel phenotype that exhibited higher CD31+ vessel counts and leaky vasculature on magnetic resonance imaging (MRI) was sensitive to VDA treatment while the PDX with the stromal vessel phenotype was resistant to therapy.	This study underlines the impact that the phenotypic and functional vascular heterogeneity of HNSCC has on the antivascular therapy in PDX models.
Rich et al., 2018 New York USA [34]	HPV+ and negative PDX models of HNSCC	Photoacoustic imaging (PAI) was utilized for longitudinal assessment of tumor hemodynamics (oxygenation saturation and hemoglobin concentration)	Consistent with PAI results, immunohistochemical staining of vascularity (CD31) and DNA damage (phosphorylated γH2AX) revealed distinct patterns of response in HPV+ and HPV- xenografts	PAI can be useful for temporal mapping in the study of tumor hemodynamics, while at the same time, the values obtained can be correlated with the level of response to radiation of HNSCC.

**Table 3 diagnostics-12-01071-t003:** Patient demographics, clinical characteristics and risk factors.

First Author Year, Country	Patient Demographics	Clinical Characteristics	Risk Factors
Strüder et al., 2021, Rostock, Germany [26]	Patients with histopathologically proven HNSCC of the oral cavity, oropharynx, hypopharynx, larynx and neck lymph node metastases	Size of the tumor >2 cm; primary disease or recurrence; >18 years of age	Smoking 73% Alcohol 58% P16 15% positive
Kang et al., 2020 Kyungbuk South Korea [29]	Oral cavity cancer	Stage IV 68%	P16 positive 84.2%
Berggren et al., 2019 Albuquerque, NM [31]	Oral cavity, oropharynx, and larynx cancers	-	-
Lilja-Fischer et al., 2019 Aarhus Denmark [32]	Patients who underwent examination under anesthesia with diagnostic biopsies or tonsillectomy on suspicion of OPSCC	-	P16 positive 70.6%
Folaron et al., 2019 NY USA [33]	HHSCC cancers Clinical stage IVA	-	P16 positive 29.5%

## Data Availability

Data are contained within the article.

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
