# Peer review of "Diagnostics of HNSCC Patients: An Analysis of Cell Lines and Patient-Derived Xenograft Models for Personalized Therapeutical Medicine"

_diagnostics, 2022, doi:10.3390/diagnostics12051071_

Round 1
Reviewer 1 Report
The authors have written a review using articles published in the last 5 years to summarise the head and neck head and neck squamous cell carcinoma targeted therapy using derived cell lines cultures and patient retrieved xenografts. In total 14 studies were included for analysis.
I would like to congratulate the authors for their effort and for their work in the field.
In the Materials and Methods chapter there is data missing regarding the exclusion and the inclusion criteria during the search.
Using a risk of bias tool to properly assess the quality of the data adds more value to the conclusions of the research. I suggest analysing the included 14 studies using a risk of bias tool.
Even though it is not a systematic review, a flow chart would be beneficial for the reader to better see the review workflow.
The tables are not correctly labelled, both are labelled as Table 3 with the same description .
Line 144: analysis for the studies
Author Response
Point-by-point response to the reviewers’ and editor’s comments
Title: “Diagnostics of HNSCC patients: an analysis of cell lines and patient-derived xenograft models for personalized therapeutical medicine”
We thank the reviewer for giving us the opportunity to improve the quality of our manuscript.
Reviewer #1:
Comments and Suggestions for Authors
The authors have written a review using articles published in the last 5 years to summarise the head and neck head and neck squamous cell carcinoma targeted therapy using derived cell lines cultures and patient retrieved xenografts. In total 14 studies were included for analysis.
I would like to congratulate the authors for their effort and for their work in the field.
- In the Materials and Methods chapter there is data missing regarding the exclusion and the inclusion criteria during the search.
- Response to Reviewer: We thank the reviewer for this comment. We have included data regarding the inclusion and exclusion criteria during the search, as suggested (Lines 163 - 172).
- Materials and Methods
Literature search and study selection: a systematic search of the PubMed and the EMBASE databases was carried out for all the published studies on HNSCC in the last 5 years, using the following search algorithm: HNSCC HPV positive PI3K positive targeted therapy, HNSCC cell lines PI3K targeted therapy, pdx xenograft HNSCC HPV PI3K targeted therapy. We performed a systematic analysis for the studies that were published in English, from the 1st of January 2017 to the 1st of March 2022, and that described and analysed the methods used for optimal targeted therapy of HNSCC patients. The 14 studies were blinded analysed by two persons. We excluded review papers and studies which have tested cell lines to other therapies except PI3K inhibitors.
- Using a risk of bias tool to properly assess the quality of the data adds more value to the conclusions of the research. I suggest analysing the included 14 studies using a risk of bias tool.
- Response to Reviewer: We thank the reviewer for this comment. We have included data regarding the inclusion and exclusion criteria during the search, as suggested. We have added the bias toll as suggested (Lines 163 - 172).
- Materials and Methods
Literature search and study selection: a systematic search of the PubMed and the EMBASE databases was carried out for all the published studies on HNSCC in the last 5 years, using the following search algorithm: HNSCC HPV positive PI3K positive targeted therapy, HNSCC cell lines PI3K targeted therapy, pdx xenograft HNSCC HPV PI3K targeted therapy. We performed a systematic analysis for the studies that were published in English, from the 1st of January 2017 to the 1st of March 2022, and that described and analysed the methods used for optimal targeted therapy of HNSCC patients. The 14 studies were blinded analysed by two persons. We excluded review papers and studies which have tested cell lines to other therapies except PI3K inhibitors.
- Even though it is not a systematic review, a flow chart would be beneficial for the reader to better see the review workflow.
- Response to Reviewer: We thank the reviewer for this comment. We added the flow chart of analyzed studies, as suggested (Lines 173 - 177).
Fig. 1: Flow chart of the analyzed studies
- The tables are not correctly labelled, both are labelled as Table 3 with the same description.
- Response to Reviewer: We thank the reviewer for this comment. We have used the correct titles and labelling for both tables, I and II, as suggested (lines 193, 210).
Table I: Diagnostic evaluation of HNSCC cell lines targeted PI3K inhibitor therapy
Table II: Diagnostic evaluation of pdx HNSCC targeted PI3K inhibitor therapy
- Line 144: analysis for thestudies
- Response to Reviewer: We thank the reviewer for this comment. We have used modified the manuscript as suggested (line 144).
Reviewer 2 Report
Comments to the authors:
- It is not clear to me the novelty or the new added scientific value of the research as there are many similar reviews in high-impact journals. authors should highlight the significance and the contribution to the scientific research?
- authors should have more details on the risk factors affecting the formation of HNSCC, environmental or genetic factors. in addition to highlighting the anatomical sites of HNSCC development
- What are the incidence, prevalence, mechanism, progression, and mortality of HNSCC
- There should be a section about the genomic alterations and key pathways involved in the formation of HNSCC and the clinical presentation (diagnosis, screening, and prevention)
- there should be a table summary of the patient demographics and clinical characteristics with the risk factors and clinical outcomes
- what are the implications of PI3K pathway alteration for EGFR pathway in HNSCC
- How can the PI3K pathway can be targeted in clinical by PI3K inhibitors, for example, buparlisib, etc
- figures and table legends should be comprehensive and detailed
- would the PDX formation be used to predict the increased risk to HNSCC
- authors should look into the molecular profiling of engrafting vs nonengrafting patient samples as been reported in the literature
- there should be a section on the preclinical assessment of CDK4 and CKD6 inhibition using abemaciclib etc in PDX models
- English language should be revised carefully
Author Response
Point-by-point response to the reviewers’ and editor’s comments
Title: “Diagnostics of HNSCC patients: an analysis of cell lines and patient-derived xenograft models for personalized therapeutical medicine”
We thank the reviewer for giving us the opportunity to improve the quality of our manuscript.
Reviewer #2:
Comments and Suggestions for Authors
Comments to the authors:
- It is not clear to me the novelty or the new added scientific value of the research as there are many similar reviews in high-impact journals. authors should highlight the significance and the contribution to the scientific research?
- Response to Reviewer: We thank the reviewer for this comment. We added the required data, as suggested (Lines 379 – 383).
Our study is a descriptive review of the recent findings regarding the latest research studies which are focusing on the evaluation of the most optimal targeted therapy of HNSCC, using cell lines and HNSCC derived pdx as models.
- Authors should have more details on the risk factors affecting the formation of HNSCC, environmental or genetic factors. In addition to highlighting the anatomical sites of HNSCC development
- Response to Reviewer: We thank the reviewer for this comment. We have added the required data as suggested (Lines 32 - 44).
HNSCC represents a heterogenous group of tumors, including cancer of the oropharynx, oral cavity, pharynx, and larynx. The recognized risk factors for HNSCC are smoking, chronic alcohol use, lack of oral hygiene, and HPV 16 in oropharyngeal cancers. Recent meta-analyses have confirmed smoking as a risk factor for HNSCC: Alotaibi M et al., found that smoking is a negative prognostic factor for overall survival in patients with hr-HPV+ [1], and Ference R et al., found that current smoking during treatment is associated with the greatest reduction in survival [2]. Interestingly, Skoulakis A et al., found in their meta-analysis that smoking is less common in HPV positive group than in HPV negative group [3]. The role of HPV in HNSCC was confirmed by a meta-analysis which included 148 studies and 12 163 cases of HNSCC from 44 countries, and the authors found HPV16 was present in more than 80% of all HPV DNA positive cases [4].
- [1] - Alotaibi, M.; Valova, V.; Hänsel, T.; Stromberger, C.; Kofla, G.; Olze, H.; Piwonski, I.; Albers, A.; Ochsenreither, S.; Coordes, A. Impact of Smoking on the Survival of Patients With High-Risk HPV-Positive HNSCC: A Meta-Analysis. In Vivo 2021, 35, 1017–1026, doi:21873/invivo.12345.
- [2]- Ference, R.; Liao, D.; Gao, Q.; Mehta, V. Impact of Smoking on Survival Outcomes in HPV-Related Oropharyngeal Carcinoma: A Meta-Analysis. Otolaryngol Head Neck Surg 2020, 163, 1114–1122, doi:1177/0194599820931803.
- [3]- Skoulakis, A.; Tsea, M.; Koltsidopoulos, P.; Lachanas, V.; Hajiioannou, J.; Petinaki, E.; Bizakis, J.; Skoulakis, C. Do Smoking and Human Papilloma Virus Have a Synergistic Role in the Development of Head and Neck Cancer? A Systematic Review and Meta-Analysis. J BUON 2020, 25, 1107–1115.
- [4] - Ndiaye, C.; Mena, M.; Alemany, L.; Arbyn, M.; Castellsagué, X.; Laporte, L.; Bosch, F.X.; de Sanjosé, S.; Trottier, H. HPV DNA, E6/E7 MRNA, and P16INK4a Detection in Head and Neck Cancers: A Systematic Review and Meta-Analysis. Lancet Oncol 2014, 15, 1319–1331, doi:1016/S1470-2045(14)70471-1.
- What are the incidence, prevalence, mechanism, progression, and mortality of HNSCC
- Response to Reviewer: We thank the reviewer for this comment. We have added the required data as suggested (Lines 42 - 60).
Updated information regarding incidence, prevalence and mortality are available on the Cancer Today website, by the International Agency for Research on Cancer (IARC). The estimated age-standardized incidence rates, in 2020, for lip, oral cavity, oropharynx, nasopharynx, hypopharynx cancers, both sexes, all ages, were 7.4 in USA, 12.7 in France, 12.9 in Romania, 6.5 in Brazil, 4.8 in China, 9.0 Namibia and 9.8 in Australia [5]. The estimated number of prevalent cases (5-year) as a proportion in 2020, hypopharynx, lip, oral cavity, nasopharynx, oropharynx, both sexes, all ages were were 39.2 in USA, 65.9 in France, 59.6 in Romania, 20.6 in Brazil, 20.1 in China, 12.0 Namibia and 51.0 in Australia [6]. The estimated age-standardized mortality rates (World) in 2020, hypopharynx, lip, oral cavity, nasopharynx, oropharynx, both sexes, all ages were 1.4 in USA, 3.0 in France, 6.7 in Romania, 3.2 in Brazil, 2.5 in China, 5.5 Namibia and 1.6 in Australia [7].
- [5] Ferlay, J.; Ervik, M.; Lam, F.; Colombet, M.; Mery, L.; Piñeros, M.; Znaor, A.; Soerjomataram, I.; Bray, F. Global Cancer Observatory: Cancer Today. Available online: http://gco.iarc.fr/today/home (accessed on 20 March 2022).
Cancer incidence and mortality data
- [6] Sung, H.; Ferlay, J.; Siegel, R.L.; Laversanne, M.; Soerjomataram, I.; Jemal, A.; Bray, F. Global Cancer Statistics 2020: GLOBOCAN Estimates of Incidence and Mortality Worldwide for 36 Cancers in 185 Countries. CA Cancer J Clin 2021, 71, 209–249, doi:3322/caac.21660.
- [7] Ferlay, J.; Colombet, M.; Soerjomataram, I.; Parkin, D.M.; Piñeros, M.; Znaor, A.; Bray, F. Cancer Statistics for the Year 2020: An Overview. Int J Cancer 2021, doi:1002/ijc.33588.
CANCER TOMORROW, another project of IARC, enables a quantification of the future cancer burden changes of new cases from 2020 to 2040, both sexes, all continents, age [0-85+]: hypopharynx 46.9 %, lip, oral cavity 41.7%, nasopharynx 37.3%, oropharynx 35.5%. For Romania, the highest changes of new cases from 2020 to 2040, Both sexes, age [0-85+] are predicted to be for lip, oral cavity (6.7%) and hypopharynx (2.8%)[8].
- [8] - Ferlay, J.; Laversanne, M.; Ervik, M.; Lam, F.; Colombet, M.; Mery, L.; Piñeros, M.; Znaor, A.; Soerjomataram, I.; Bray, F. Global Cancer Observatory: Cancer Tomorrow Available online: https://gco.iarc.fr/tomorrow/en (accessed on 15 April 2022).
A recent multicenter study concluded that in some populations in the United States, more than 90% of OPSCCs are produced by HPV [9].
- [9] - Scott-Wittenborn, N.; D’Souza, G.; Tewari, S.; Rooper, L.; Troy, T.; Drake, V.; Bigelow, E.O.; Windon, M.J.; Ryan, W.R.; Ha, P.K.; et al. Prevalence of Human Papillomavirus in Head and Neck Cancers at Tertiary Care Centers in the United States over Time. Cancer 2022, 128, 1767–1774, doi:1002/cncr.34124.
- There should be a section about the genomic alterations and key pathways involved in the formation of HNSCC and the clinical presentation (diagnosis, screening, and prevention)
- Response to Reviewer: We thank the reviewer for this comment. We have added the required scientific data, suggested (Lines 290 - 305).
Genomic alterations and key pathways involved in the formation of HNSCC and the clinical presentation
Recently, many groups of researchers are focusing on the identification of new possible therapeutic targets in HNSCC. Chen M et al., 2021, identified four novel HNSCC susceptibility loci (CDKN1C rs452338, CDK4 rs2072052, E2F2 rs3820028 and E2F2 rs2075993) as genetic alterations in the cell cycle pathway are common in HNSCC [57]. Zhang Z et al., analysed for the first time the 2-hydroxyisobutylated modification proteomic for OSCC, which is significantly concentrated in the actin cytoskeleton regulatory pathway, suggesting that this pathway may mediate the oncogenesis or exacerbation of OSCC [58]. Li Y et al., studied the role and molecular mechanism of cyclin-dependent kinase 5 (CDK5) in regulating the growth of tongue squamous cell carcinoma (TSCC). The authors found that increased levels of CDK5 expression in TSCC tissues was established as an independent risk factor affecting TSCC growth and patient prognosis. CDK5 was proven as an oncogene in TSCC and is considered as a molecular marker for use in the diagnosis and treatment of TSCC [59]. Another research team identified a promising therapeutic target and prognosis marker for human OSCC: Actin-like protein 8, which was detected to play an oncogenic role in the pathogenesis of OSCC [60]
- [57] - Chen, M.; Xu, W.-M.; Wang, G.-Y.; Hou, Y.-X.; Tian, T.-T.; Li, Y.-Q.; Qi, H.-J.; Zhou, M.; Kong, W.-J.; Lu, M.-X. Genetic Variants of Cell Cycle Pathway Genes Are Associated with Head and Neck Squamous Cell Carcinoma in the Chinese Population. Carcinogenesis 2021, 42, 1337–1346, doi:10.1093/carcin/bgab094.
- [58] - Zhang, Z.; Xie, H.; Zuo, W.; Tang, J.; Zeng, Z.; Cai, W.; Lai, L.; Lu, Y.; Shen, L.; Dong, X.; et al. Lysine 2-Hydroxyisobutyrylation Proteomics Reveals Protein Modification Alteration in the Actin Cytoskeleton Pathway of Oral Squamous Cell Carcinoma. J Proteomics 2021, 249, 104371, doi:10.1016/j.jprot.2021.104371.
- [59] - Li, Y.; Yao, F.; Jiao, Z.; Su, X.; Wu, T.; Peng, J.; Yang, Z.; Chen, W.; Yang, A. Cyclin-Dependent Kinase 5 Promotes the Growth of Tongue Squamous Cell Carcinoma through the MicroRNA 513c-5p/Cell Division Cycle 25B Pathway and Is Associated with a Poor Prognosis. Cancer 2022, 128, 1775–1786, doi:10.1002/cncr.34136.
- [60] - Wang, L.; Xing, X.; Tian, H.; Fan, Q. Actin-like Protein 8, a Member of Cancer/Testis Antigens, Supports the Aggressive Development of Oral Squamous Cell Carcinoma Cells via Activating Cell Cycle Signaling. Tissue Cell 2022, 75, 101708, doi:10.1016/j.tice.2021.101708.
Screening and prevention of HNSCC (Lines 306 – 317).
Many research studies are focusing on identification of the optimal biomarker for early detection of HNSCC, as the routine clinical evaluation include clinical examination and radiological assessment.
Huang L et al., identified in a meta-analysis that circular RNAs showed high accuracy in the diagnosis of OSCC and could be used as prospective biomarkers for optimal diagnostic [61].
Another possible biomarker for early detection is hypermethylated DNA in saliva and oral swabs for OSCC, which proved to have optimal accuracy and raised hope for optimization of fast detection of these cancers’ evolution, given its noninvasive sampling procedures [62].
Gaw G et al., supported non-invasive hypermethylation markers using saliva and oral swabs for OSCC diagnosis, together with other biomarkers, to optimize sensitivity and specificity of screening [63].
One possibility to prevent HPV-associated HNSCC is HPV vaccination programme implementation, for males also [64].
- [61] - Huang, L.; Pei, T.; Wu, G.; Liu, J.; Pan, W.; Pan, X. Circular RNAs as a Diagnostic Biomarker in Oral Squamous Cell Carcinoma: A Meta-Analysis. J Oral Maxillofac Surg 2022, 80, 756–766, doi:10.1016/j.joms.2021.11.021.
- [62] -Adeoye, J.; Alade, A.A.; Zhu, W.-Y.; Wang, W.; Choi, S.-W.; Thomson, P. Efficacy of Hypermethylated DNA Biomarkers in Saliva and Oral Swabs for Oral Cancer Diagnosis: Systematic Review and Meta-Analysis. Oral Dis 2022, 28, 541–558, doi:10.1111/odi.13773.
- [63] - Gaw, G.; Gribben, M. Can We Detect Biomarkers of Oral Squamous Cell Carcinoma from Saliva or Mouth Swabs? Evid Based Dent 2022, 23, 32–33, doi:10.1038/s41432-022-0248-9.
- [64] - Palacios, V.J.; Merlino, D.J.; Anderson, S.S.; Yeakel, S.R.; Choby G, G.W.; Wiedermann, J.P.; Moore, E.J.; O’Byrne, T.J.; Jacobson, R.M.; Van Abel, K.M. Feasibility of Instituting a Clinical Otolaryngology Human Papillomavirus (HPV) Vaccination Program. Laryngoscope 2022, doi:10.1002/lary.30130.
- there should be a table summary of the patient demographics and clinical characteristics with the risk factors and clinical outcomes
- Response to Reviewer: We thank the reviewer for this comment. We added the suggested for the available data in the analyzed studies (Table III, lines 615 - 617).
Table III: Patient demographics and clinical characteristics and risk factors
|
First Author Year, Country |
Patient demographics |
Clinical Characteristics |
Risk Factors |
|
Strüder D et al., 2021, Rostock, Germany [26] |
Patients with histopathologically proven HNSCC of the oral cavity, oropharynx, hypopharynx, larynx and neck lymph node metastases;
|
size of the tumor > 2 cm; primary disease or recurrence; > 18 years of age. |
Smoking 73 % Alcohol 58 % P16 15% positive |
|
Kang HN et al., 2020 Kyungbuk South Korea [29] |
Oral cavity cancer |
Stage IV 68 % |
P16 positive 84,2 %
|
|
Berggren KL et al., 2019 Albuquerque, NM [31] |
Oral cavity, oropharynx, and larynx cancers |
- |
- |
|
Lilja-Fischer JK et al., 2019 Aarhus Denmark [32] |
patients who underwent examination under anesthesia with diagnostic biopsies or tonsillectomy on suspicion of OPSCC. |
- |
P16 positive 70.6 % |
|
Folaron M et al, 2019 NY USA [33] |
HHSCC cancers Clinical stage IVA |
- |
P16 positive 29.5% |
- what are the implications of PI3K pathway alteration for EGFR pathway in HNSCC
- Response to Reviewer: We thank the reviewer for this comment. We have added the required scientific data, suggested (Lines 318 - 327).
Many studies are analysing the implications of PI3K pathway alteration for EGFR pathway in HNSCC.
Zaryouh H et al., are considering that co-targeting EGFR and the PI3K/Akt pathway could have a synergistic drug effect, improving sensitivity to EGFR and clinical efficacy. Optimal selection is important, for patients which could benefit from this targeted therapy [65].
Anti-EGFR monoclonal antibody cetuximab was evaluated in recurrent and/or metastatic HNSCC patients in a phase I dose-escalation trial, and the authors stopped the trial because unfavourable toxicity profile [18].
Genetic alterations in Epidermal growth factor receptor (EGFR) and PI3K are common in HNSCC: Mock A et al., found that more than half of HPV-negative HNSCC showed a pathway activation in EGFR or PI3K [66].
- [65] - Zaryouh, H.; De Pauw, I.; Baysal, H.; Peeters, M.; Vermorken, J.B.; Lardon, F.; Wouters, A. Recent Insights in the PI3K/Akt Pathway as a Promising Therapeutic Target in Combination with EGFR-Targeting Agents to Treat Head and Neck Squamous Cell Carcinoma. Med Res Rev 2022, 42, 112–155, doi:1002/med.21806.
- [18] - Marret, G.; Isambert, N.; Rezai, K.; Gal, J.; Saada-Bouzid, E.; Rolland, F.; Chausson, M.; Borcoman, E.; Alt, M.; Klijanienko, J.; et al. Phase I Trial of Copanlisib, a Selective PI3K Inhibitor, in Combination with Cetuximab in Patients with Recurrent and/or Metastatic Head and Neck Squamous Cell Carcinoma. Invest New Drugs 2021, 39, 1641–1648, doi:1007/s10637-021-01152-z.
- [66] - Mock, A.; Plath, M.; Moratin, J.; Tapken, M.J.; Jäger, D.; Krauss, J.; Fröhling, S.; Hess, J.; Zaoui, K. EGFR and PI3K Pathway Activities Might Guide Drug Repurposing in HPV-Negative Head and Neck Cancers. Front Oncol 2021, 11, 678966, doi:3389/fonc.2021.678966.
- How can the PI3K pathway can be targeted in clinical by PI3K inhibitors, for example, buparlisib, etc
- Response to Reviewer: We thank the reviewer for this comment. We have added the required scientific data, suggested (Lines 328 - 341).
A clinical trial published in Lancet Oncology by Soulières D et al., evaluated if the addition of buparlisib to paclitaxel could optimize clinical outcomes compared with paclitaxel and placebo in patients with recurrent or metastatic HNSCC. After 2 years of follow up of more than 150 patients, the authors observed improved clinical efficacy with a manageable safety profile, and they suggested that buparlisib in combination with paclitaxel could be an effective second-line treatment for recurrent or metastatic HNSCC patients [67].
One year later, the same research team published another clinical trial in which patients with TP53 alterations, HPV-negative status, low mutational load had a better overall survival from combination therapy of buparlisib and paclitaxel [68].
Brisson RJ et al., evaluated on 12 patients the maximum tolerated dose of buparlisib given concurrently with cetuximab in recurrent and metastatic HNSCC. In this pilot study, the authors concluded that buparlisib at 100 mg daily plus cetuximab proved to be well-tolerated [15].
Lenze N et al., considered that buparlisib, a class I pan-PI3K inhibitor, with an optimal tolerated toxicity, could improve the actual 5-year overall survival for HNSCC, which is only at around 50-66% [69].
- [67] - Soulières, D.; Faivre, S.; Mesía, R.; Remenár, É.; Li, S.-H.; Karpenko, A.; Dechaphunkul, A.; Ochsenreither, S.; Kiss, L.A.; Lin, J.-C.; et al. Buparlisib and Paclitaxel in Patients with Platinum-Pretreated Recurrent or Metastatic Squamous Cell Carcinoma of the Head and Neck (BERIL-1): A Randomised, Double-Blind, Placebo-Controlled Phase 2 Trial. Lancet Oncol 2017, 18, 323–335, doi:1016/S1470-2045(17)30064-5.
- [68] - Soulières, D.; Licitra, L.; Mesía, R.; Remenár, É.; Li, S.-H.; Karpenko, A.; Chol, M.; Wang, Y.A.; Solovieff, N.; Bourdeau, L.; et al. Molecular Alterations and Buparlisib Efficacy in Patients with Squamous Cell Carcinoma of the Head and Neck: Biomarker Analysis from BERIL-1. Clin Cancer Res 2018, 24, 2505–2516, doi:1158/1078-0432.CCR-17-2644.
- [15] - Brisson, R.J.; Kochanny, S.; Arshad, S.; Dekker, A.; DeSouza, J.A.; Saloura, V.; Vokes, E.E.; Seiwert, T.Y. A Pilot Study of the Pan-Class I PI3K Inhibitor Buparlisib in Combination with Cetuximab in Patients with Recurrent or Metastatic Head and Neck Cancer. Head Neck 2019, 41, 3842–3849, doi:1002/hed.25910.
- [69] - Lenze, N.; Chera, B.; Sheth, S. An Evaluation of Buparlisib for the Treatment of Head and Neck Squamous Cell Carcinoma. Expert Opin Pharmacother 2021, 22, 135–144, doi:1080/14656566.2020.1825684.
- figures and table legends should be comprehensive and detailed
- Response to Reviewer: We thank the reviewer for this comment. We have used the correct titles and labelling for both tables, I and II, as suggested (lines 193, 210).
Table I: Diagnostic evaluation of HNSCC cell lines targeted PI3K inhibitor therapy
Table II: Diagnostic evaluation of pdx HNSCC targeted PI3K inhibitor therapy
- would the PDX formation be used to predict the increased risk to HNSCC
- Response to Reviewer: We thank the reviewer for this comment. We have used the correct name of the virus in all the manuscript as suggested (Lines 343 - 357).
Some group researchers have used PDX formation to predict the increased risk to HNSCC.
Facompre ND et al. established well-characterized PDXs and organoids from HPV+ HNSCCs and they proved the retaining of PIK3CA mutations, TRAF3 deletion and the absence of EGFR amplifications, NOTCH1 mutations. The authors have identified, in a PDX model, reduced E7 and p16INK4A levels, which were associated with recurrent HPV+ HNSCCs and with lethal outcome. For prediction of disease recurrence risk, the authors have also analysed on PDX the E2F target gene expression, as a useful biomarker [27].
PDX models have been used to predict the risk to HNSCC, and in evaluation of potential biomarkers, such as Remodelin, an inhibitor of NAT10 (one of the most promising prognostic risk gene). In their study, the authors demonstrated it significantly suppressed the growth of HNSCC in a PDX model, indicating that Remodelin may be an optimal candidate drug for HNSCC treatment [70].
In a very recent study, the authors aimed and succeeded to developed a prediction score for locoregional failure and distant metastases in OSCC that incorporates PDX engraftment, beside the known clinicopathological risk factors [71].
- [27] - Facompre, N.D.; Rajagopalan, P.; Sahu, V.; Pearson, A.T.; Montone, K.T.; James, C.D.; Gleber-Netto, F.O.; Weinstein, G.S.; Jalaly, J.; Lin, A.; et al. Identifying Predictors of HPV-Related Head and Neck Squamous Cell Carcinoma Progression and Survival through Patient-Derived Models. Int J Cancer 2020, 147, 3236–3249, doi:1002/ijc.33125.
- [70] - Tao, W.; Tian, G.; Xu, S.; Li, J.; Zhang, Z.; Li, J. NAT10 as a Potential Prognostic Biomarker and Therapeutic Target for HNSCC. Cancer Cell Int 2021, 21, 413, doi:1186/s12935-021-02124-2..
- [71] - Id Said, B.; Ailles, L.; Karamboulas, C.; Meens, J.; Huang, S.H.; Xu, W.; Keshavarzi, S.; Bratman, S.V.; Cho, B.C.J.; Giuliani, M.; et al. Development and Validation of an Oral Cavity Cancer Outcomes Prediction Score Incorporating Patient-Derived Xenograft Engraftment. JAMA Otolaryngol Head Neck Surg 2022, 148, 342–349, doi:1001/jamaoto.2022.0003.
- authors should look into the molecular profiling of engrafting vs nonengrafting patient samples as been reported in the literature
- Response to Reviewer: We thank the reviewer for this comment. We followed the suggestion of the reviewer, and we added the data from literature regarding molecular profiling in engrafting vs non-engrafting PDX (Lines 358 - 369).
In a comprehensive study, Karamboulas C. et al. have studied the molecular profiling of 64 engrafters and 48 non-engrafters, which were tested for DNA mutations or number of copies alterations with a custom hybrid capture targeted sequencing panel of 112 genes. The authors found no statistically significant associations between any single gene mutation and engraftment. They identified that CDKN2A mutation or deletion, a CCND1 amplification, or both were present in 83.4% in case of rapid engrafters, and in only 18,2% in case of nonengrafting / slow engrafters, wich suggest that genomic deregulation of the G1/S checkpoint pathway correlates with engraftment [72].
Using Illumina Cancer Hotspot Panel, Strüder D et al., analysed the molecular profile (TP53, KDR, KRAS, SMARCB1, EGFR) of 13 patient samples and corresponding PDX, and they found that molecular pathology is preserved in PDX, with the mention regarding the importance of intratumoral heterogeneity [26].
- [72] - Karamboulas, C.; Bruce, J.P.; Hope, A.J.; Meens, J.; Huang, S.H.; Erdmann, N.; Hyatt, E.; Pereira, K.; Goldstein, D.P.; Weinreb, I.; et al. Patient-Derived Xenografts for Prognostication and Personalized Treatment for Head and Neck Squamous Cell Carcinoma. Cell Rep 2018, 25, 1318-1331.e4, doi:1016/j.celrep.2018.10.004.
- [26] Strüder, D.; Momper, T.; Irmscher, N.; Krause, M.; Liese, J.; Schraven, S.; Zimpfer, A.; Zonnur, S.; Burmeister, A.-S.; Schneider, B.; et al. Establishment and Characterization of Patient-Derived Head and Neck Cancer Models from Surgical Specimens and Endoscopic Biopsies. J Exp Clin Cancer Res 2021, 40, 246, doi:1186/s13046-021-02047-w.
- there should be a section on the preclinical assessment of CDK4 and CKD6 inhibition using abemaciclib etc in PDX models
- Response to Reviewer: We thank the reviewer for this comment. We have added the section as suggested (Lines 370 - 378).
An impressive multidisciplinary research team from Princess Margaret Cancer Centre, Toronto, Canada, raised the level of using PDX models from recapitulating many of the features of their corresponding clinical cancers, including histopathological and molecular profiles, to preclinical assessment of CDK4 and CKD6 inhibition, using abemaciclib, on a large collection of 243 HPV negative HNSCC patient-derived xenograft models. The authors are underlying the necessity of using this type of CDK4 and CDK6 inhibitors in case of HNSCC patients with the poorest prognosis, and even more, they are mentioning the necessity of further studies for identification of mechanisms of resistance of abemaciclibin, alone or in association with ionizing radiation [72].
- [72] - Karamboulas, C.; Bruce, J.P.; Hope, A.J.; Meens, J.; Huang, S.H.; Erdmann, N.; Hyatt, E.; Pereira, K.; Goldstein, D.P.; Weinreb, I.; et al. Patient-Derived Xenografts for Prognostication and Personalized Treatment for Head and Neck Squamous Cell Carcinoma. Cell Rep 2018, 25, 1318-1331.e4, doi:1016/j.celrep.2018.10.004.
- English language should be revised carefully
- Response to Reviewer: We thank the reviewer for this comment. We have revised English language as suggested.

Round 2
Reviewer 1 Report
We would like to thank the authors for the modifications made as we suggested.
Reviewer 2 Report
No further comments